# Comparative Study of Sensory and Physicochemical Characteristics of Green-Tea-Fortified Cupcakes upon Air Frying and Oven Baking

**DOI:** 10.3390/foods12061266

**Published:** 2023-03-16

**Authors:** Hiu-Lok Ngan, Shu-Yu Ip, Mingfu Wang, Qian Zhou

**Affiliations:** 1Shenzhen Key Laboratory of Marine Microbiome Engineering, Institute for Advanced Study, Shenzhen University, Shenzhen 518060, China; 2School of Biological Sciences, The University of Hong Kong, Hong Kong 999077, China

**Keywords:** air fryer, household oven, cupcake model, green tea catechins (GTCs), advanced glycation end products (AGEs)

## Abstract

The air fryer and the oven are common cooking methods in our daily lives. However, previous investigations of the air fryer were limited to its comparison with deep-fat frying. This study compared the differences between air frying and household oven baking (without a fan or other forced airflow inside) on food quality and physicochemical properties using a cupcake model. Results showed that the oven-baked cupcakes were softer in texture (87.15%), greener in color (6.07%), and lower in weight loss (7.78%) and toxic advanced glycation end products (AGEs, 21.40%) when the heating temperature and duration were the same as oven baking. To improve the sensory characteristics and health value, the cupcakes were fortified with green tea. The differences in texture, color, and level of toxicants between the two cooking methods were diminished after the addition of green tea. Moreover, the chemical profiles of green tea catechins in the green-tea-fortified cupcakes remained similar upon thermal cooking, except that the air-fried cupcakes were lower in gallic acid (GA) but higher in (−)-gallocatechin (GC). Collectively, based on the differences in heating mechanisms, our data indicated that oven baking is a better cooking method suitable to prepare cupcakes than air frying from the perspectives of sensory characteristics and food safety, while green tea additives effectively counter the drawbacks of the air fryer.

## 1. Introduction

Thermal cooking methods, including baking and frying, are essential for food preparation in both the food industry and home cooking. The term baking is preferred to describe the preparation of dough products, and it is defined as “cooking food in an oven in air to which water vapor may or may not be added”. Frying is to dehydrate food through a heat transfer medium, such as air, fat, or oil, in direct contact with the food [1]. Among a variety of frying techniques, air frying technology was developed and became popular for the household preparation of fried foods to reduce the use of oil and to achieve a similar product quality [2,3]. The air fryer is mainly based on using the circulation of hot air to cook foods. Hence, due to their similarity in thermodynamic properties, many previous studies compared the differences between air frying and deep-fat frying in microstructure and sensory properties by using starch-based food models, including doughnuts and French fries. Particularly, Gouyo et al. pointed out that deep-fat-fried French fries were crispier than air-fried samples, which was possibly owing to the difference in the pore diameter and pore size distribution in the crust. Ghaitaranpour et al. found that air-fried doughnuts were smoother and softer on the surface than deep-fat-fried samples [4,5,6]. However, air frying is different from deep-fat frying from the perspective of food engineering. The air fryer intrinsically is an oven-designed cooker to improve the circulating airflow problem of the conventional oven [3]. By modifying the shape of the heating chamber, air and food materials are forced to contact with the enhanced air velocity in the air fryer than in the conventional oven [7]. In the conventional oven, foods are cooked only by the heat source (the heat pipes), which has no fan or other forced airflow apparatus inside. To date, only a few works have been reported comparing air frying with oven baking [8]. To be specific, the comparisons were merely limited to unraveling the impact of airflow on the product volume in a conventional oven by using starch-based models such as bread and moist cakes [9,10]. Hence, a comprehensive evaluation of the product quality of air-fried food and a comparison with oven-baked food is needed.

Food additives are commonly used to improve the nutritional value, sensory characteristics, and shelf-life of foods. Agricultural products or their extracts, which contain a high level of polyphenols, have been used as natural additives in bakery products [11]. Thereinto, green tea is a potential candidate due to its beneficial effects such as anti-carcinogenicity, cardioprotective ability, neuroprotective ability, and its attenuating effect on Parkinson’s disease [12,13]. In addition, green tea and its extracts are widely added to a variety of cooking foods to offer a characteristic flavor and additional health benefits. The active components in green tea were identified as caffeine (CAF) and catechins (GTCs). These GTCs included (−)-epigallocatechin-3-gallate (EGCG), (+)-catechin (CAT), (−)-epicatechin-3-gallate (ECG), (−)-epigallocatechin (EGC), (−)-gallocatechin-3-gallate (GCG), (−)-epicatechin (EC), (−)-catechin-3-gallate (CG), and (−)-gallocatechin (GC), with EGCG being the most abundant component [14,15]. GTCs are known as strong radical scavengers and inhibitors against the formation of some harmful Maillard reaction products (MRPs) [16,17]. For example, EGCG can trap 3-oxo propanamide to prevent the formation of carcinogen acrylamide in several food systems [18,19,20] and scavenge reactive di-carbonyl species (RCS) to inhibit the formation of pro-inflammatory advanced glycation end products (AGEs) in bakery foods [21,22]. AGEs are a group of heterogenous compounds that are formed at the late stage of the Maillard reaction during thermal processing. Abundant in sugar, proteins, and oils, cupcakes tend to form toxic AGEs because of reactions that start from the carbonyl group of reducing sugar and the amino group of proteins and oils [17]. GTCs were reported to show potential for the development of safe food products [16]. Therefore, it is still unknown whether under the two cooking methods, air frying and oven baking, GTCs can have different effects on the formation of harmful MRPs or not.

The present research first compared the sensory and physicochemical characteristics of cupcakes prepared by using air frying and oven baking. The impacts of green-tea-fortified cupcakes on these characteristics were then examined for both cooking techniques. Considering that AGEs are the predominant MRPs in starch-based foods, AGEs were chosen as representative hazardous substances, and their levels in air-fried and oven-baked cupcakes were detected. Finally, to understand the effects of GTCs, we evaluated their thermal stability upon air frying and oven baking separately.

## 2. Materials and Methods

### 2.1. Chemicals and Reagents

Premium Green Tea (Imperial Choice, Shenzhen, China), white cake mix (Betty Crocker TM, Minneapolis, MN, USA), sunflower oil (Yu Pin King^®^, Hong Kong, China), and brown eggs (Yu Pin King^®^, Hong Kong, China) were purchased from a local supermarket in Hong Kong. CAT, EGCG, ECG, EGC, and GCG were obtained from Cool Chemical Technology Co., Ltd. (Beijing, China). EC, CG, GC, CAF, GA, Tris-HCl, and formic acid were purchased from Sigma-Aldrich (Saint Louis, MO, USA). Tween-20, sodium dodecyl sulfate (SDS), and 2-mercaptoethanol were obtained from Bio-Rad Laboratories (Hercules, CA, USA). Methanol and hydrochloric acid in AR grade were purchased from RCI Labscan Limited (Bangkok, Thailand). Acetonitrile in HPLC grade was purchased from Merck KGaA (Darmstadt, Germany).

### 2.2. Preparation of Green Tea Powder and Extract

Dried green tea leaves (Premium Green Tea) were blended into powder form for their fortification use in cupcakes and were safely edible in our experimental section of “sensory evaluation”. After that, green tea extract (in methanol) was prepared to evaluate the effect of green tea on the physicochemical characteristics of cupcakes. Briefly, the extract was prepared by extracting 90 g of the powder twice with 900 mL of methanol for 30 min through sonication at room temperature. Another 200 mL of methanol was used for rinsing during filtration. To determine the amount of each green tea component, a few volumes of green tea extract in methanol were sampled, diluted in methanol, and filtered through a 0.45 µm nylon syringe filter prior to HPLC analysis by using the external calibration method [23]. Following that, the green tea extract was concentrated under vacuum at 35 ± 1 °C to reach 2 different concentration levels, 0.058 ± 0.001 g (Level 1) and 0.094 ± 0.001 g (Level 2) of EGCG per gram of green tea extract. The concentration of each level of green tea extract was confirmed by triplicate sampling prior to HPLC analysis. All extracted solutions were stored in an air-tight screw bottle at −20 °C and rewarmed by sonication to room temperature for 5 min before use.

### 2.3. Preparation of Cupcakes

The cupcake mix was prepared according to the instructions printed on the external packaging material of the commercial cake premix (Betty CrockerTM Super MoistTM Favorites White Cake Mix) with minor modifications. In brief, a cup of water (237.00 g), a half-cup of sunflower oil (118.50 g), and 3 eggs were mixed with the premix powder (461.00 g). In each batch, 6 cupcakes (20.00 g each) were prepared. After mixing, each 20.00 g of cupcake mix was transferred into individual aluminum cups and weighed before thermal treatment. Both upper and lower heating elements of the oven (Hauswirt HO-30C, with the nominated power of 1600 watts) and the air fryer (Philips HD9743/11, with the nominated power of 1400 watts) were preheated concurrently at 180 °C for 15 min. The thermal condition was set to 180 °C for 10 min for both machines. After heating, the air-fired and oven-baked cupcakes were cooled at room temperature for 30 min. Physical measurements for color, weight, and firmness were conducted. Following that, the cupcake samples were ground and stored in a −20 °C freezer until use.

For the green-tea-fortified cupcakes, two levels were prepared based on the concentration of EGCG added to the cupcakes. In each batch of 6 cupcakes, 6.00 g of Level 1 green tea extract or Level 2 green tea extract (prepared in Section 2.2) was added with 120.00 g of the cupcake mix for 2 min, respectively. The final concentration of EGCG was 0.058 g in each cupcake mix for the low level of green-tea-fortified cupcakes (LGCs) and was 0.094 g in each cupcake mix for the high level of green-tea-fortified cupcakes (HGCs). Moreover, for the analysis of EGCG-fortified cupcakes, pure EGCG was added, and its concentration was 0.094 g EGCG in each cupcake. Other procedures were the same as mentioned above in the non-green-tea-added cupcakes.

### 2.4. Sensory Evaluation of Cupcakes

Ground green tea powder with the equivalent amount of EGCG as in the green tea extract was alternatively used to replace the extract in the case of any safety issues. Quantitative descriptive analysis (QDA) was used to evaluate the air-fried and oven-baked cupcakes with and without green tea powder fortification. Six attributes (i.e., sweetness, bitterness, oiliness, astringency, aroma, and doneness) were selected. For each attribute, 10 intensity anchors were set as the integrals between and included 0 and 10 (0 = dislike very much, 5 = intermediate, 10 = like very much). Ten panelists (6 females and 4 males, aged 20–30) were recruited and each panelist needed to evaluate 6 samples prepared by different food preparation methods, including control air-fried cupcake, control oven-baked cupcake, air-fried LGC, oven-baked LGC, air-fried HGC, and oven-baked HGC. Cupcakes were assessed sequentially from the cupcake with the least to the most astringency. Drinking water was supplied to rinse the mouth between samples. Each cupcake was assessed for the attributes of its appearance first, followed by the assessment of its attributes of taste.

### 2.5. Color Measurement of Cupcakes

Three chromatic cylindrical coordinates (*L**, *a**, and *b**) of the ground cupcake samples were measured by a colorimeter (CR-400, Konica Minolta, Japan). For each sample, measurements were performed for 4 times and recorded as *L**, *a**, and *b** values, where *L** reflects the brightness of the color (0 = black, 100 = white), *a** indicates redness/greenness (−*a** = greenness, *a** = redness), and *b** implies blueness/yellowness (−*b** = blueness, *b** = yellowness). The E-index of each cupcake sample was calculated by the following equation, and the difference of 2 E-indices (∆E) was used to compare the color distance between 2 samples [24].
$$E=\sqrt{{L*}^{2}+{a*}^{2}+{b*}^{2}}$$

### 2.6. Firmness Measurement of Cupcakes

The firmness of the freshly prepared cupcakes was measured by using a TA-XT plus Texture Analyzer with the Texture Exponent software version 2.0.7.0 (Stable Micro-systems, Godalming, UK). The test speed was 1 mm/s, and the press distance was 5 mm. A trigger force of 5× *g* was enforced on the entire standing cupcake by a round probe with a diameter of 36 mm (P/36).

### 2.7. Weight Loss Determination of Cupcakes

Weight loss of cupcakes upon thermal processing was determined by the percentage decrease in weight that a whole batch of cupcake before heating to after heating and cooling.

### 2.8. Determination of Fluorescent AGEs in Cupcakes

A 100 mL volume of extraction buffer was prepared by dissolving 0.05 g of Tween-20, 1.0 g of SDS, 5.0 g of 2-mercaptoethanol, and 50 mM Tris-HCl (pH 7.4) into 100 mL of water. The pH value of the buffer was adjusted to pH 7.40 by adding hydrochloric acid (37%). Following that, each 1.00 g of ground cupcake sample was mixed with 4.75 mL of the extraction buffer, vortexed for 30 s, and mechanically shaken at 150 rpm/min at room temperature overnight to extract the fluorescent AGEs. After shaking, the cupcake samples were centrifuged at 2020× *g* for 15 min to precipitate large particles. Then, 1 mL of the aqueous supernatant was further centrifugated at 14,000× *g* for 5 min, and each 200 µL of supernatant was pipetted into a black 96-well plate. Finally, the total fluorescent AGEs were measured by using a Victor X4 Multilabel Plate Reader (PerkinElmer, Waltham, MA, USA). The excitation and emission wavelengths adopted in this study were 360/40 nm and 460/40 nm, respectively [25].

### 2.9. Chromatographic Analysis of Green Tea Components

The active components in the green-tea-fortified cupcakes were extracted by the method developed by the Weibiao Zhou Group with some modifications and then subjected to HPLC analysis [26]. Briefly, 4.00 g of ground cupcake sample was extracted by 40 mL of methanol at room temperature for 60 min by a mechanical shaker operating at 150 rpm/min. After shaking, 5 mL of methanol extract was then centrifuged at 2020× *g* for 15 min. The supernatant was filtered at 0.45 µm of nylon syringe filter and diluted by methanol according to the ratio of weight loss upon heating. A reverse-phase Shimadzu HPLC system (LC-20AT) was applied, with a photodiode array detector (SPD-M20A) and a Phenomenex Prodigy C18 column (4.6 × 250 mm, 5 μm; Torrance, CA, USA). The mobile phases were 0.2% formic acid in water (A) and acetonitrile (B), and the gradient elution program was as follows: 0 min, 7% B; 16 min, 12% B; 25 min, 22% B; 33 min, 27% B; hold 3 min; 39 min, 40% B; back to 7% B in 1 min and held for 5 min. The flow rate was 0.8 mL/min, and the injection volume was 10 µL.

To analyze green tea components, an 11-mixed standard stock solution (2 mM) was prepared by dissolving 0.03763 g of GA, 0.06125 g of GC and EGC, 0.03884 g of CAF, 0.05805 g of C and EC, 0.09167 g of EGCG and GCG, and 0.08847 g of ECG and CG into 100 mL of methanol and storing in a −20 °C freezer until use. All working solutions were prepared daily by serial dilution in methanol. The concentrations of the standard working solutions were 0.5, 1, 2, 5, 10, 50, and 100 µM for each component except EGCG. The calibration curve of EGCG involved the high-point calibration up to 200 µM, and its low-point calibration was down to 1 µM. The calibration curves of each component were established corresponding to their wavelength with maximum absorption (271 nm for GA, 270 nm for GC and EGC, 272 nm for CAF, 278 nm for C and EC, 273 nm for EGCG, 274 nm for GCG, 276 nm for ECG and CG). The calibration curves were linear with correlation coefficients (*R*^2^) of at least 0.9993.

To determine the most principal component in the green tea-fortified cupcakes responsible for the variations during thermal treatment, a data table of the aligned peaks was imported into partial least squares–discriminant analysis (PLS-DA) using SIMCA-P version 13.0 software package (Umetrics, Umeå, Sweden). All variables were mean-centered and scaled to Pareto variance, and the variable importance for the projection (VIP) was used to spot the chemical marker in the green tea-fortified cupcakes.

### 2.10. Statistical Analysis

All descriptive statistics were computed with the GraphPad Prism 5 software package (GraphPad Software Inc., La Jolla, CA, USA) and Microsoft Excel 2016. Student’s t-test was used to assess statistically significant differences, by which a value of *p* < 0.05 was considered statistically significant. Data are expressed as mean ± standard deviation (SD) of at least triplicate determinations in this study.

## 3. Results and Discussion

### 3.1. The Impact of Air Frying and Oven Baking on the Characteristics of Cupcakes

As shown in Figure 1A, the air fryer caused irregular bumps on the surface of the cupcakes. The volume expansion of the cupcakes during air frying creates a more porous structure with larger voids [9]; therefore, the highly porous microstructure might differentiate in appearance between the ground air-fried cupcake samples and the oven-baked samples. The lower uniformity of the air-fried cupcakes might be due to the presence of the forced hot airflow during heating because a higher volume expansion could be observed when the heat penetrated the cupcakes [10]. A sensory evaluation study of the air-fried and oven-baked cupcakes was conducted to understand consumers’ acceptability of the taste of the cupcakes prepared by the two methods. The average intensity value of each attribute was used to draw a radar chart (Figure 1B). The results showed that the oven-baked cupcakes were higher in doneness, lower in oiliness, and almost the same as air-fried cupcakes in sweetness, bitterness, and astringency (Figure 1B). Sweetness, bitterness, and astringency seemed to be the intrinsic properties of cakes, which may only rely on the ingredients. However, no significance could be observed between air frying and oven baking for all six attributes.

After the sensory evaluation, a variety of physical measurements (including color, firmness, and weight loss) were performed to further characterize the food qualities of the oven-baked and air-fried cupcakes. To begin with, *L**, *a**, and *b** values were used to measure and describe the color change of the cupcakes in this study. As illustrated in Figure 1C–E, the oven-baked cupcakes were slightly higher in brightness (*p* = 0.1800 in L value), significantly lower in redness (*p* = 0.0020 in a value), and slightly higher in yellowness (*p* = 0.7463 in b value). The calculated E values were 51.00 ± 0.38 and 52.61 ± 0.52 for air frying and oven baking, respectively, which were significantly different (Figure 1F). Color distances were described by the ∆E value, which was 1.61 between the two cooking methods. Furthermore, the firmness of the cupcakes was determined by a texture analyzer. We found that air frying produced cupcakes that were 6.78-fold higher in firmness (Figure 1G). This indicated that the air fryer made harder cupcakes than the oven, and the panelists seemed to prefer the softer texture. In addition, the weight loss of cupcakes upon thermal treatments was observed and was determined by percentage loss of weight (Figure 1H). Our results indicated that air frying and oven baking resulted in 22.2 ± 0.2% and 20.5 ± 0.3% weight loss, respectively. Hence, air frying led to a significantly greater weight loss than oven baking (*p* = 0.0024). This finding implies that the heating mechanism of the air fryer should differ from that of the conventional oven, although both cookers implement hot air to transfer thermal energy. The higher weight loss was owing to the higher loss in water. This finding also reinforced that the heating with airflow in the air fryer exhibits higher efficiency in heat supply [9,10], suggesting that a shorter heating duration should be adopted in its household application compared to the heating time setting of conventional ovens.

Several lines of evidence have exemplified that starch-based foods are prone to produce harmful substances, especially AGEs and acrylamide generated by the Maillard reaction during thermal treatment. For instance, the formation of fluorescent AGEs has been found in cookies upon oven baking. Continuous consumption of AGEs has been proven to induce chronic diseases, including cardiovascular diseases, senile dementia, and malignant tumors [25,27]. Therefore, declining the level of these hazardous substances in baking foods is critical to human health. Intriguingly, a lower acrylamide content in French fries was reported by using oven baking compared to air frying [28]. Dry heat such as in baking and frying is known to promote 10–100-fold dietary AGE formation [29]. Thus, in this study, we investigated the extent of AGE formation upon thermal treatment and found that air frying was significantly higher (13.3%) in fluorescent AGE formation than oven baking when the heating temperature and duration settings were the same (*p* = 0.0021, Figure 1I). AGE accumulation in the human body was reported to be involved in the onset/progression/propagation of diabetic complications [27]. However, the study on the association between the daily tolerance of dietary AGE intake and the occurrence of these pathological conditions was still lacking. Future work can contribute to this field. Therefore, our results implied that oven baking might be generally healthier than air frying to prepare starch-based food products.

### 3.2. Green Tea Fortification Reduced the Characteristic Differences between Oven-Baked and Air-Fried Cupcakes

Generally speaking, our sensory evaluation data tended to imply that oven baking can offer a better taste to green-tea-fortified cupcakes (Figure 2A). To be specific, oven-baked cupcakes were higher in doneness, bitterness, and astringency, while sweetness and oiliness were almost the same as air-fried cupcakes in both LGCs and HGCs. Again, no significant differences were found. On the other hand, our results also showed that the average intensities in both air-fried and oven-baked cupcakes decreased along with the increment in the concentration of green tea additives. This might be because bitterness and astringency were positively associated with the content of polyphenolics presented in the cupcakes. Meanwhile, it seemed that in LGCs, oven baking and air frying caused more diversity in bitterness (Figure 2A), which had little difference in the non-fortified cupcakes (Figure 1B). Therefore, current data were not able to give any explanation of this phenomenon, and further efforts are needed. Moreover, the addition of green tea generally reduced the oily mouthfeel. This might have contributed to the lipophilicity of polyphenols.

The physical parameters of cupcakes that characterized their resultant food product quality (*L**, *a**, and *b** values and firmness) were compared upon air frying and oven baking. To be specific, green tea fortification lowered the *L**, *a**, and *b** values of the cupcakes along with the increment in green tea concentration, presenting a more tea-like color (Figure 2B–D). Therefore, no significant color change could be observed between oven baking and air frying, either in the LGC- or the HGC-added group. However, on the basis of the fundamental data of cupcakes (non-green-tea-added samples) in Figure 1C–E, green-tea-fortified cupcakes were lower in darkness and redness, while higher in yellowness. Our results agreed with the work by the Andrzej Półtorak Group, in which the same observations were reported in green-tea-fortified sponge-fat cakes [30]. This is probably due to the color of green tea itself. For the two thermal processing methods, less color difference was observed when a higher concentration of green tea fortification was applied (Figure 2E). Interestingly, the firmness of air-fried cupcakes was reduced by 84.7% and 85.4% in LGCs and HGCs (compared to the non-green-tea-added samples), respectively, whereas it remained almost unchanged in the oven-baked samples (Figure 1G and Figure 2F). This shows that the heated air circulation in the air fryer causes the larger cell diameter of porous size in cakes, while the fortification of green tea can shorten the cell diameter. Our data were supported by a previous study, which also presented a decrease in the cell diameter of porous size in the central slices of bread after the addition of green tea [31]. Remarkably, the firmness of the air-fried cupcakes was comparable to that of the oven-baked cupcakes upon natural green tea addition, suggesting that the food quality of cupcakes can be effectively enhanced by green tea fortification.

In terms of hazardous substances, the content of fluorescent AGEs in the air-fried cupcakes was reduced by 43.0% and 51.7% in the LGC and HGC groups, respectively, while it decreased by 27.7% and 34.1%, respectively, in the oven-baked samples (Figure 1J and Figure 2G). By referring to literature data, GTCs (especially EGCG) are known to prevent AGE formation by trapping intermediate precursors and via antioxidation [27]. Although air frying exerted a higher risk of harmful substance formation, green tea fortification exhibited its antiglycative ability and removed the differences between the two cooking methods. Collectively, we presented that green tea supplementation in cakes can offer a healthy profile and effectively reduce the inappropriate impact of air frying.

### 3.3. The Change of Green Tea Components in Cupcakes upon Air Frying and Oven Baking

To further understand the action mechanism, comparative studies of the change of green tea components in LGCs and HGCs were performed upon air frying and oven baking. To begin with, the chemical profile of the green tea extract was elucidated by reverse-phase HPLC analysis, and the chromatogram of green tea extract (in methanol) was illustrated in Figure 3A. Our results showed that EGCG was the most abundant component in green tea, which agreed with previous literature data [32,33,34]. Furthermore, the detected GTCs included GC (retention time = 12.97 min), EGC (retention time = 7.98 min), CAT (retention time = 20.08 min), EC (retention time = 26.82 min), EGCG (retention time = 27.71 min), GCG (retention time = 28.96 min), and ECG (retention time = 34.45 min). CG was undetected in the green tea extract before its addition to the cupcake mix. As illustrated in the chromatograms in Figure 3A–C, EC, EGCG, and ECG extracted from the heated cupcakes were lower in amount than those in green tea extract, while the amounts of CAT, GCG, and CG were increased. A similar tendency was described in literature data that EC, EGCG, and ECG were found to be lowered, and GCG and CG were boosted when they were heated in solution form. It was estimated that epimerization of GTCs from their epi-forms to non-epi-forms occurred spontaneously under thermal conditions in the water- or moist-containing cupcake model [35]. Therefore, the air-fried and oven-baked cupcakes showed no difference regarding the chemical profiles. Our PLS-DA results indicated that inter-group variations existed among the cupcakes fortified with different concentrations of green tea extract (Figure 4A). VIP analysis found that EGCG was the predominant chemical marker in the four groups of green-tea-fortified cupcakes (Figure 4B). Therefore, EGCG was selected to perform a single-component cupcake model, and the results demonstrated that besides the conversion to GCG, EGCG was degraded into GA (Figure 3D–E). The degradation of epi-forms of GTCs into GA was also suggested by the increment of GA upon thermal treatment in Yuerong Liang Group’s study [35]. Hence, it can be concluded that simultaneous epimerization and degradation were the major reactions causing the reduction of EGCG.

The levels of green tea components from the two different matrices were evaluated individually. As presented in Figure 5, in the LGC groups, EC (Figure 5F), EGCG (Figure 5G), and ECG (Figure 5I) extracted from the oven-baked cupcakes were lower in amount than those from the air-fried cupcakes, while the amount of CAT (Figure 5E), GCG (Figure 5H), and CG (Figure 5J) were higher. Although these results had no significant difference, they implied that the cupcakes in the conventional oven suffered from a higher degree of epimerization. Moreover, in the HGC groups, the oven-baked samples were significantly higher in GA (Figure 5A) but significantly lower in GC than the air-fried samples (Figure 5B), indicating that the cupcakes in the conventional oven also underwent a higher degree of degradation. We proposed the higher extent of epimerization and degradation was due to the lower weight loss during oven baking, retaining a higher volume of water or moisture to maintain the cake mix in its solution form. Collectively, our current data indicated that air frying and oven baking had little difference in retaining GTCs during thermal treatment.

## 4. Conclusions

In the present study, we compared air frying with conventional oven baking from the perspectives of sensory and physicochemical properties by using a cupcake model. Results showed that the air-fried cupcakes were higher in weight loss, lower in conformity, significantly higher in firmness, slightly darker in color, and significantly higher in redness and in AGE formation than oven baking when the heating temperature and duration time were the same as the settings of the conventional oven. Our data further suggested that the addition of green tea modified the physical characteristics of the air-fried and oven-baked cupcakes significantly, and it removed the differences between the two cooking methods. Moreover, no significant change in green tea profile was found upon air frying and oven baking, except that GA was lower and GC was higher in air-fried HGCs. In addition, to the best of our knowledge, this study is the first to compare air frying with oven baking in household machines by using a natural-additive-fortified starch-based food model. We believe our work may not only draw attention to the safety of household foods, but also provide insights into the development of functional foods. However, several improvements can be proposed for future studies in this field, including (1) the optimized heating duration setting of air frying should be specifically investigated for each bakery product, and (2) more sensitive analytical instrumentation such as HPLC/MS/MS should be applied to perform chemical analysis of other potential harmful MRPs, such as polycyclic aromatic hydrocarbons.

## Figures and Tables

**Figure 1 foods-12-01266-f001:**
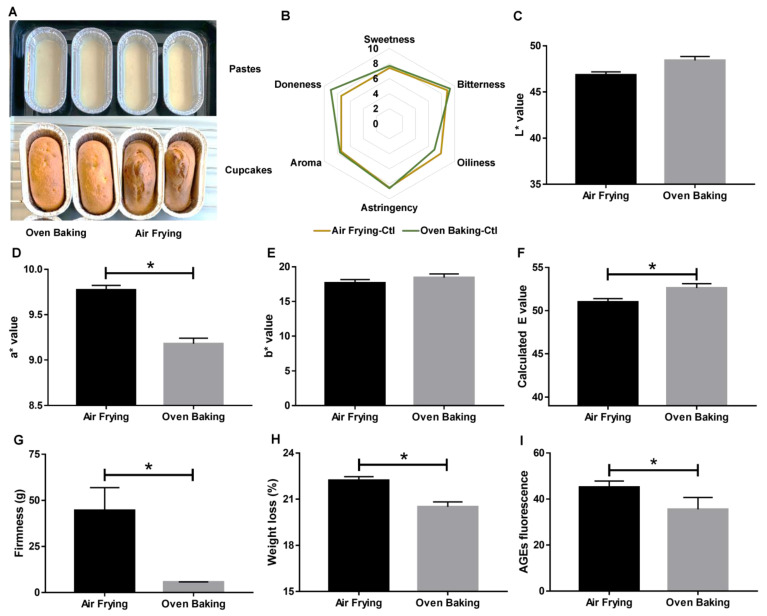
Characteristic comparisons between air-fried and oven-baked cupcakes. (**A**) Represented picture of cupcakes. (**B**) Radar chat of sensory evaluation. (**C**–**E**) Lab value. (**F**) Calculated E value. (**G**) Firmness. (**H**) Weight loss. (**I**) AGE fluorescence. * *p* < 0.05.

**Figure 2 foods-12-01266-f002:**
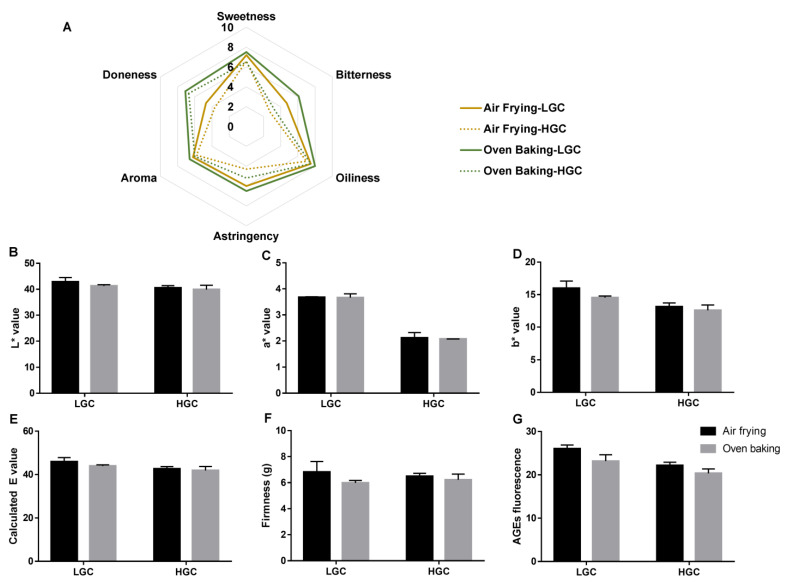
Green tea fortification removed the difference between air-fried and oven-baked cupcakes. (**A**) Radar chart of sensory evaluation. (**B**–**D**) Lab value. (**E**) Calculated E value. (**F**) Firmness. (**G**) AGE fluorescence. LGC, the low level of green-tea-fortified cupcake; HGC, the high level of green-tea-fortified cupcake.

**Figure 3 foods-12-01266-f003:**
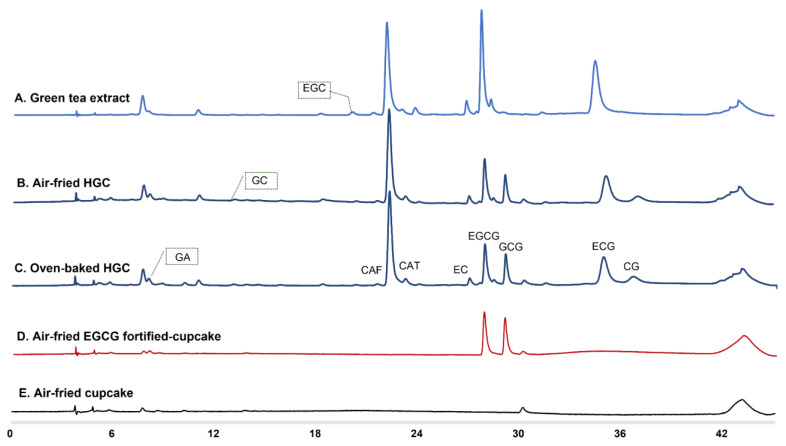
Chromatographic analysis of green tea-fortified cupcakes. (**A**–**C**) HPLC chromatography of GTCs in the green extract, air-fried HGCs, and oven-baked HGCs. (**D**) HPLC chromatography of GTCs in air-fried EGCG-fortified cupcakes. A 94 mg amount EGCG was added to each 20 g of cupcake mix. (**E**) HPLC chromatography of the control air-fried cupcakes. The wavelength was set at 273 nm and the addition level of green tea extract was Level 2 (94 mg of EGCG in 20 g of cupcake mix). GA, gallic acid; GC, (−)-gallocatechin; EGC, (−)-epigallocatechin; CAF, caffeine; CAT, (+)-catechin; EC, (−)-epicatechin; EGCG, (−)-epigallocatechin-3-gallate; GCG, (−)-gallocatechin-3-gallate; ECG, (−)-epicatechin-3-gallate; CG, (−)-catechin-3-gallate.

**Figure 4 foods-12-01266-f004:**
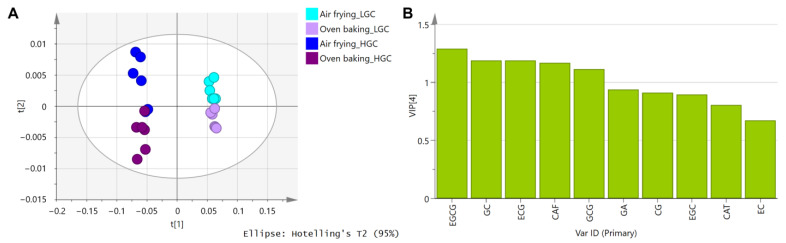
Partial least squares–discriminant analysis (PLS-DA) of air-fried and oven-baked cupcakes. (**A**) Inter-group variations of air-fried and oven-baked cupcakes. (**B**) Variable importance (VIP) of the green tea components. LGC, the low level of green-tea-fortified cupcake; HGC, the high level of green-tea-fortified cupcake; EGCG, (−)-epigallocatechin-3-gallate; GC, (−)-gallocatechin; ECG, (−)-epicatechin-3-gallate; CAF, caffeine; GCG, (−)-gallocatechin-3-gallate; GA, gallic acid; CG, (−)-catechin-3-gallate; EGC, (−)-epigallocatechin; CAT, (+)-catechin; EC, (−)-epicatechin.

**Figure 5 foods-12-01266-f005:**
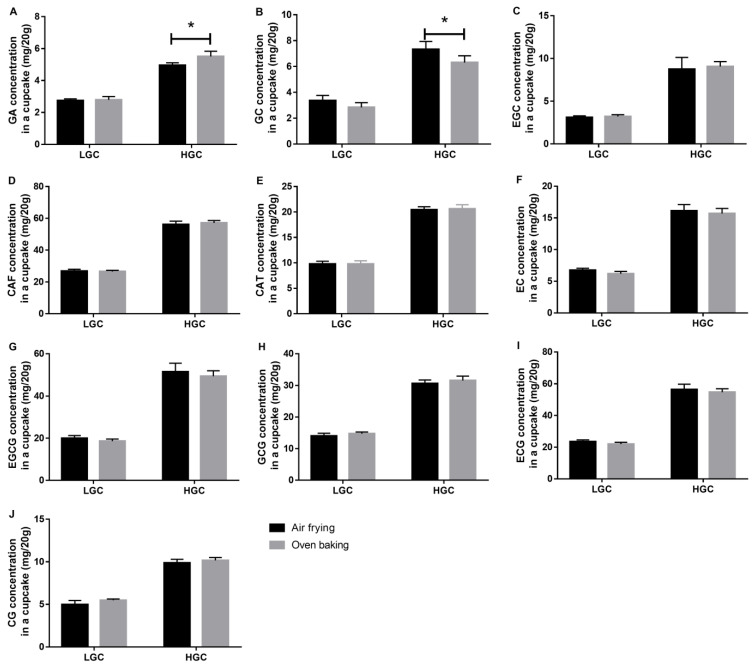
Comparison of GTCs and caffeine in air-fried and oven-baked cupcakes. (**A**–**J**) Comparison of GA, GC, EGC, CAF, CAT, EC, EG, GCG, ECG, and CG, respectively. * *p* < 0.05. LGC, the low level of green-tea-fortified cupcake; HGC, the high level of green-tea-fortified cupcake; GA, gallic acid; GC, (−)-gallocatechin; EGC, (−)-epigallocatechin; CAF, caffeine; CAT, (+)-catechin; EC, (−)-epicatechin; EGCG, (−)-epigallocatechin-3-gallate; GCG, (−)-gallocatechin-3-gallate; ECG, (−)-epicatechin-3-gallate; CG, (−)-catechin-3-gallate.

## Data Availability

The data used to support the findings of this study are available from the corresponding author upon request.

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
