# Peer review of "Comparative Study of Sensory and Physicochemical Characteristics of Green-Tea-Fortified Cupcakes upon Air Frying and Oven Baking"

_foods, 2023, doi:10.3390/foods12061266_

Round 1

Reviewer 1 Report (Previous Reviewer 1)

The quality of the manuscript was improved after the requested modifications.

Author Response

Thank you so much for your positive reply!

Reviewer 2 Report (Previous Reviewer 2)

I checked the comments of the previous reviewers and the revisions made by the authors

The authors were kind enough to respond

Author Response

Thank you so much for your positive reply!

Reviewer 3 Report (New Reviewer)

This manuscript evaluated green tea-fortified cupcakes' sensory and physicochemical characteristics upon air frying and oven baking. The topic is exciting and the manuscript is well-designed. It needs a minor revision.

1. The preparation of cupcakes designed based on commercial products or common recipes?

2. L 334;  "* p <0.05" should be removed and shifted to figure 5 (L 391).

3. For better understanding, mention the full phrase of abbreviations under the figures. 

Author Response

This manuscript is a resubmission of an earlier submission. The following is a list of the peer review reports and author responses from that submission.

Round 1

Reviewer 1 Report

I am very grateful you for the invitation to review the manuscript foods-2124409 by Ngan and coauthors "Comparative study of sensory and physicochemical characteristics of green tea-fortified cupcakes upon air frying and oven baking”. This study compared air frying with conventional oven baking on food quality and physiochemical properties using a cupcake model. The work is interesting but needs adjustments to increase the quality of the material.

Comments:

- Abstract: Insert advantages of this type of process. The authors must detail if this is not similar to the convection cooking process. Highlight differences in operation and indicate the possibility of expanding the scale, for example. In addition, information about the addition of green tea is not presented (advantages, reasons, among others).

- Abstract: Highlight whether conventional oven baking refers to the conduction oven.

- Abstract, Lines 12-15: Authors must present numerical results of the most significant results.

- Line 14: Indicate what heat-induced toxicants are.

- Line 15: “green tea fortification”: This objective is not mentioned before.

- Line 16: Specify which determinations this refers to: “eliminated the differences between the two cooking methods”

- Lines 16-19: This objective is not mentioned before.

- Abstract: The abstract must be fully revised to make the content clearer and more accurate.

- Lone 19, “oven baking is more feasible to cupcakes than air frying”: Please indicate the thermodynamic, and physical fundamentals related to this.

- Lines 21-22: Change the repeated keywords by different words from the title.

- Lines 25-44: The comparison with the oil frying method makes no sense in the introduction. The comparison of the work is between “upon air frying and oven baking”. A theoretical discussion should be carried out on the mechanisms of these two techniques. Cupcake preparation traditionally involves baking in an oven, not frying in oil.

- Introduction: The thermodynamic and physical principles of the two techniques must be presented.

- Introduction: Despite the indication of antioxidant components, it was not mentioned how green tea can participate in the improvement of technological properties since an improvement in the characteristics was presented after its addition (comparing the two techniques).

- Lines 67-70: Authors should include a brief discussion on the formation of substances by the techniques used. Review the indication during changes to the introduction.

- Lines 83-84: Indicate in the sentence “for sensory evaluation” what the authors refer to. Wasn't the sensory only for the cupcake?

- Line 84, “After that, green tea extract was used for evaluating the effect of GTCs”: What effect? Do the authors refer to identify and quantify? It is not clear what was done.

- Revise the minutes abbreviation throughout the text.

- Line 108, “Cupcake samples were then ground and stored in -20 ºC freezer until use”: I believe that before that the samples had been analyzed in relation to the physical aspects, at least.

- Line 127, “Each panelist”: Specify the number of evaluators.

- Line 147-148: This is not methodological content. Consider removal or transfer to a suitable item.

- 3. Results and Discussion: Please consider the same sequence presented in the material and methods item during the presentation of results and discussion.

- Lines 208-209: Sensory evaluation is usually carried out in later stages to evaluate the characteristics of the evaluated product in relation to the physicochemical and technological properties.

- Lines 211-212: Include the information in the methodology item.

- Lines 245-247: Only in this sentence is there an indication that the “conventional” oven can be convective. This needs to be made clearer throughout the work.

- Lines 250-251: Please move and add the sentence before the figure.

- Figure 1 and 2 (A and B): Please use different colors for the appearance (A) and taste (B) attributes of sensory evaluation.

- Lines 265-268: It is unclear the risks associated with these identified levels. Authors must inform available legal parameters, compare with other products, etc.

- Lines 269-270: The item title is unconventional for a scientific article. Please readjust the title.

- There is no discussion about the tea components that influence the color of the cupcake.

- Lines 315-316: Please move and add the sentence before the figure.

- Discussion: Therefore, authors should improve the discussion and add comparative reports to other works, including variations, and legal limits, among others.

Author Response

We would like to thank the reviewer He/She offered many useful suggestions for us to improve the quality of our manuscript. We truly value the opinions. Thanks indeed!

Response to Reviewer 1 Comments

Point 1: Abstract: Insert advantages of this type of process. The authors must detail if this is not similar to the convection cooking process. Highlight differences in operation and indicate the possibility of expanding the scale, for example. In addition, information about the addition of green tea is not presented (advantages, reasons, among others).

Response 1: We have revised the abstract accordingly and clearly mentined in Line 11-12 that the oven is not convectional. More explanations are added in the Introduction part (Line 34-35 and Line 43-45). The purpose of add green tea is added in Line 17.

Point 2: Abstract: Highlight whether conventional oven baking refers to the conduction oven.

Response 2: The oven we used was not conduction oven, which didn’t have a fan inside to induce convection of hot air. To avoid misunderstanding, wer have clearly mentioned this in Line 11-12.

Point 3: Abstract, Lines 12-15: Authors must present numerical results of the most significant results.

Response 3: The numerical results is added in Line 15-16.

Point 4: Line 14: Indicate what heat-induced toxicants are.

Response 4: It is revised in accrodingly Line 15-16.

Point 5: Line 15: “green tea fortification”: This objective is not mentioned before.

Response 5: It is revised as “To improve the sensory characteristics and health value, cupcakes were fortified with green tea” in Line 17-18.

Point 6: Line 16: Specify which determinations this refers to: “eliminated the differences between the two cooking methods”

Response 6: We have replaced the sentence by “The differences in texture, colour, and level of toxicants were diminished between the two cooking methods after the addition of green tea” in Line 18-19.

Point 7: Lines 16-19: This objective is not mentioned before.

Response 7: It is revised accordingly in Line 17-18.

Point 8: Abstract: The abstract must be fully revised to make the content clearer and more accurate.

Response 8: We have re-writen the Abstract accordingly in Line 10-23.

Point 9: Line 19, “oven baking is more feasible to cupcakes than air frying”: Please indicate the thermodynamic, and physical fundamentals related to this.

Response 9: We have rephrased the sentence as “Collectively, basing on the differences in heating mechanisms, our data indicated that oven baking is more feasible to cupcakes than air frying from the perspectives of sensory characteristics and food safety, …” in Line 21-24.

Point 10: Lines 21-22: Change the repeated keywords by different words from the title.

Response 10: We have replaced the keywords “air frying”, “oven baking”, and “cupcakes” by “air fryer”, “household oven”, and “cupcake model”, respectively.

Point 11: Lines 25-44: The comparison with the oil frying method makes no sense in the introduction. The comparison of the work is between “upon air frying and oven baking”. A theoretical discussion should be carried out on the mechanisms of these two techniques. Cupcake preparation traditionally involves baking in an oven, not frying in oil.

Response 11: We have revised the paragraph in Line 29-50. Also, we want to explain that definition of the term “frying” is necessary to introduce the concept of frying through air. This is the reason why we kept the sentence “Frying is to dehydrate food through a heat transfer medium, such as air, fat, or oil, in direct contact with the food”. Besides that, mentioning the previous research spotted on is needed to claim that there is a necessity on comparing air frying with oven baking rather than comparing with deep-fat frying, because air is the heating medium for both air fryer and conventional oven. We sincerely hope you may consider our opinions! If not, we can rephrase it in the next submmision.

Point 12: Introduction: The thermodynamic and physical principles of the two techniques must be presented.

Response 12: We have revised it accordingly in Line 36-38 and Line 46-50.

Point 13: Introduction: Despite the indication of antioxidant components, it was not mentioned how green tea can participate in the improvement of technological properties since an improvement in the characteristics was presented after its addition (comparing the two techniques).

Response 13: We have explained how green tea prevented toxic AGEs in Line 339-340. But for the improving of other characteristics, there’s still lack of sufficient literature data especially findings from the molecular level.

Point 14: Lines 67-70: Authors should include a brief discussion on the formation of substances by the techniques used. Review the indication during changes to the introduction.

Response 14: We have revised it accordingly in Line 71-75 and Line 79-85.

Point 15: Lines 83-84: Indicate in the sentence “for sensory evaluation” what the authors refer to. Wasn't the sensory only for the cupcake?

Response 15: We have revised it accordingly in Line 99-101.

Point 16: Line 84, “After that, green tea extract was used for evaluating the effect of GTCs”: What effect? Do the authors refer to identify and quantify? It is not clear what was done.

Response 16: We have revised it accordingly in Line 101-102.

Point 17: Revise the minutes abbreviation throughout the text.

Response 17: We have revised “mins” into “min” throughout the text.

Point 18: Line 108, “Cupcake samples were then ground and stored in -20 ºC freezer until use”: I believe that before that the samples had been analyzed in relation to the physical aspects, at least.

Response 18: We have revised it accordingly in Line 125-127. Sorry, we did a careless mistake here.

Point 19: Line 127, “Each panelist”: Specify the number of evaluators.

Response 19: We have revised it accordingly in Line 146-147.

Point 20: Line 147-148: This is not methodological content. Consider removal or transfer to a suitable item.

Response 20: We have deleted the sentences accordingly.

Point 21: 3. Results and Discussion: Please consider the same sequence presented in the material and methods item during the presentation of results and discussion.

Response 21: We have exchanged the positions of “Firmness measurement of cupcakes” and “Weight loss determination of cupcakes” in the section of “Materials and Methods” to make the sequence the same.

Point 22: Lines 208-209: Sensory evaluation is usually carried out in later stages to evaluate the characteristics of the evaluated product in relation to the physicochemical and technological properties.

Response 22: We partly agree with you that sensory evaluation is usually carried out in later stages. But in our study, we treated sensory evaluation as a preliminary study and we want to know its acceptability in the beginning. If the acceptability was too low, we might tend to use other functional ingredients.

Point 23: Lines 211-212: Include the information in the methodology item.

Response 23: We have revised it accordingly in Line 146-147.

Point 24: Lines 245-247: Only in this sentence is there an indication that the “conventional” oven can be convective. This needs to be made clearer throughout the work.

Response 24: We have revise it accordingly in the whole manuscript.

Point 25: Lines 250-251: Please move and add the sentence before the figure.

Response 25: We have revised it accordingly for the figures.

Point 26: Figure 1 and 2 (A and B): Please use different colors for the appearance (A) and taste (B) attributes of sensory evaluation.

Response 26: We think it might be easier to be understood if the same sample use the same colour.

Point 27: Lines 265-268: It is unclear the risks associated with these identified levels. Authors must inform available legal parameters, compare with other products, etc.

Response 27: We have revised it accordingly in Line 286-291.

Point 28: Lines 269-270: The item title is unconventional for a scientific article. Please readjust the title.

Response 28: We have revised it accordingly in Line 292-293.

Point 29: There is no discussion about the tea components that influence the color of the cupcake.

Response 29: We have revised it accordingly in Line 321-324.

Point 30: Lines 315-316: Please move and add the sentence before the figure.

Response 30: We have revised it accordingly for the figures.

Point 31: Discussion: Therefore, authors should improve the discussion and add comparative reports to other works, including variations, and legal limits, among others.

Response 31: We have tried our best to reviese the discussions accordingly.

Reviewer 2 Report

This article with title " Comparative study of sensory and physicochemical characteristics of green tea-fortified cupcakes upon air frying and oven baking"is useful for food industrial.  The tables and figures were comprehensive and appropriate.

 It would be better if the tests related to antioxidant activity,  lipid oxidation (peroxide value, free fatty acid and Thiobarbituric acid index) and physicochemical parameters (protein, fat, carbohydarate and ash) in cupcakes were checked. The article was well written and had valuable scientific information. In the introduction should be added the relevant articles, especially the previous articles of the authors mentioned in the abstract and Line 272: remove this extra “in”.

Author Response

Thanks a lot for your suggestions! We have tried out best to revise them.

Response to Reviewer 2 Comments

Point 1: This article with title " Comparative study of sensory and physicochemical characteristics of green tea-fortified cupcakes upon air frying and oven baking"is useful for food industrial.  The tables and figures were comprehensive and appropriate.

It would be better if the tests related to antioxidant activity, lipid oxidation (peroxide value, free fatty acid and Thiobarbituric acid index) and physicochemical parameters (protein, fat, carbohydarate and ash) in cupcakes were checked. The article was well written and had valuable scientific information. In the introduction should be added the relevant articles, especially the previous articles of the authors mentioned in the abstract and Line 272: remove this extra “in”.

Response 1: Thanks a lot for the comments! Our cupcake project is still ongoing, the antioxidant activity, lipid oxidation and other physicochemical parameters are the focus of our next manuscript. Also, sorry to say we are currently not able to offer these data in such a short time. The intraduction was revised accordingly in Line 39-42. And ‘in’ was removed in Line 295.

Reviewer 3 Report

REVIEWER COMMENTS

1. The aim of the study is insufficient. Results could have been given more clearly in the abstract.

2. The abstract should be revised and rewritten in a more understandable way.

3. The introduction should be developed and the purpose should be clearly written.

4. The purpose of the study should be stated more clearly.

5. How many panelists evaluated the samples sensorial? This number can be showed in materials methods section.

6. Discussions are inadequate and not clear enough. Evaluation should be done in conjunction with current studies.

Revisions requested to be made in the article were shown in the manuscript. The manuscript can be revised.

Author Response

Thanks for your opinions! We have tried our best to revise them.

Response to Reviewer 3 Comments

Point 1: The aim of the study is insufficient. Results could have been given more clearly in the abstract.

Response 1: The abstract was revised accordingly in Line 10-24. .

Point 2: The abstract should be revised and rewritten in a more understandable way.

Response 2: The abstract has been revised accordingly in this submission.

Point 3: The introduction should be developed and the purpose should be clearly written.

Response 3: The introduction has been revised accordingly and the purpose was clealy mentioned in Line 79-85.

Point 4: The purpose of the study should be stated more clearly.

Response 4: The purpose has been clearly mentioned in the abstract (Line 11-13, 17-18) and introduction (Line 79-85).

Point 5: How many panelists evaluated the samples sensorial? This number can be showed in materials methods section.

Response 5: It is reviese accordingly in Line 146-147.

Point 6: Discussions are inadequate and not clear enough. Evaluation should be done in conjunction with current studies.

Response 6: The discussions have been revised accordingly and cojunction with other studies was added (Reference 29, 30, and 35).

Point 7: Revisions requested to be made in the article were shown in the manuscript. The manuscript can be revised.

Response 7: We have tried our best to revise the whole manuscript accordingly. We are very much obliged to your comments!

Round 2

Reviewer 1 Report

The revised version submitted by the authors does not indicate the location of the changes made in the updated document, not allowing identification of the requested changes.

Reviewer 3 Report

The wanted revisions were made by authors. The manuscript can be accepted.